

# A large transient multi-scenario multi-model ensemble of future streamflow and groundwater projections in France

Eric Sauquet[1], Guillaume Evin[2], Sonia Siauve[3], Ryma Aissat[4], Patrick Arnaud[5], Maud Bérel[6], Jérémie Bonneau[1,7], Flora Branger[1], Yvan Caballero[8], François Colléoni[5], Agnès Ducharne[9], Joël Gailhard[10], Florence Habets[11], Frédéric Hendrickx[12], Louis Héraut[1], Benoît Hingray[2], Peng Huang[9], Tristan Jaouen[1], Alexis Jeantet[13], Sandra Lanini[8], Matthieu Le Lay[10], Claire Magand[14], Louise Mimeau[1], Céline Monteil[12], Simon Munier[13], Charles Perrin[15], Olivier Robelin[1,8], Fabienne Rousset[17], Jean-Michel Soubeyroux[17], Laurent Strohmenger[15], Guillaume Thirel[15,16], Flore Tocquer[17], Yves Tramblay[18], Jean-Pierre Vergnes[4], Jean-Philippe Vidal[1]

[1]UR RiverLy, INRAE, Villeurbanne, France
[2]Univ. Grenoble Alpes, INRAE, CNRS, IRD, Grenoble INP, IGE, Grenoble, France
[3]OiEau, 87100 Limoges, France
[4]BRGM – French Geological Survey, Orléans, France
[5]UMR RECOVER, INRAE, Aix-Marseille University, Le Tholonet, France
[6]MTEECPR, La Défense, France
[7]INSA Lyon, DEEP, UR 7429, Villeurbanne, France
[8]UMR 183 G-Eau, INRAE, CIRAD, IRD, AgroParisTech, Institut Agro, BRGM, Montpellier
[9]Sorbonne Université/CNRS/EPHE, METIS-IPSL, Paris, France
[10]Département Eau Environnement, EDF-DTG, Saint Martin le Vinoux, France
[11]Geology Laboratory of Ecole Normale Supérieure, Pierre Simon Laplace Research University, CNRS UMR 8538, Paris, France
[12]Département LNHE, EDF-R&D, 78401 Chatou, France
[13]CNRM, Université de Toulouse, Météo-France, CNRS, Toulouse, France
[14]OFB, Direction de la recherche et de l'appui scientifique, Nantes, France
[15]Université Paris-Saclay, INRAE, UR HYCAR, Antony, France
[16]Univ Toulouse, CNES/IRD/CNRS/INRAE, CESBIO, Toulouse, France
[17]Météo-France, Direction de la Climatologie et des Services Climatiques, Toulouse, France
[18]UMR Espace Dev (Univ. Montpellier, IRD), Montpellier, France

*Correspondence to*: Eric Sauquet (eric.sauquet@inrae.fr)

**Abstract.** A large transient multi-scenario and multi-model ensemble of future streamflow and groundwater projections in France developed in a national project named Explore2 was recently made available. The main objective of Explore2 is to provide rich and spatially-consistent information for the future evolution of hydrological (surface and groundwater) resources and extremes in France to support adaptation strategies. The Explore2 dataset was obtained using a nested multi-scenario multi-model approach to estimate future uncertainty and to assess local climate at the catchment scale: three greenhouse gas (GHG) emission scenarios, a set of 17 combinations of Global Climate Models and Regional Climate Models (GCM-RCM), and two bias correction methods provide the meteorological forcing for nine surface hydrology models and four groundwater



hydrology models (one to simulate groundwater recharge and three to simulate groundwater level). In this paper, we present the methodology underlying the dataset, the evaluation of the hydrological models against daily streamflow and groundwater

level observations, the assessment of the future streamflow and recharge projections, the data availability and the ways of accessing the data and understanding the results (mainly through visualisation tools).

This large set of hydrological projections shows a high model agreement on the decrease in seasonal flows in the South of France under the RCP8.5 high-emission scenario, confirming its hotspot status. The surface HMs agree on the decrease in summer flows across France under the RCP8.5 scenario, with the exception of northern part France. This area may indeed

benefit from more active winter recharge that may counterbalance decrease in summer precipitation and increase in evapotranspiration. In the mountainous areas, winter flows will increase as a result of higher air temperature and the high degree of agreement between the models holds regardless of the RCP considered. Unsurprisingly, the higher the GHG emission scenario, the higher the median changes. Most of these changes are organised in France along a north-south gradient, regardless of the RCP considered.

**1. Introduction**

**1.1 From Explore2070 to Explore2**

The national Explore2070 project (2010-2013) (Carroget *et al.*, 2017) was a pioneering study in France dedicated to the assessment of the possible impacts of future climate and socio-economic changes on the streamflow and underground water in mainland France and overseas departments for the period 2046-2065. The Explore2070 climate change study was based on

seven Global Climate Models (GCMs) forced by the median greenhouse gas emission scenario A1B of the 4[th] IPCC report (2007, https://www.ipcc.ch/report/ar4/syr). In the early 2020s, the Explore2070 project was still a reference for French water managers almost 10 years after the publication of its conclusions. While the Explore2070 dataset has helped stakeholders in developing regional adaptation strategies, some limitations motivated the development of an updated dataset. Indeed, Explore2070 was restricted to the 2046-2065 period, which limited the identification of the climate response of modelling

chains because of the likely large noise added by internal variability to projections (e.g. Hingray *et al.*, 2019). In addition, other time periods (typically the end of the century for designing hydraulic infrastructures) might be of interest for diverse users. Finally, the number of points where future flows were simulated was only 1522, which is not sufficient to provide an adequate coverage of the territory, and small catchments were under-represented. Although the projections provided by Explore2070 have contributed to the development of water management strategies, the use of the produced data has also been

limited for two reasons: the data were not easily accessible (no national portal where hydrological projections can be freely downloaded), and support guidelines were not available, meaning that only stakeholders with the capacity to process and interpret highly technical data used them.

The availability of an updated and state-of-the-art ensemble of climate projections since the publication of Explore2070 conclusions, and the increased needs of users underpinned the idea of a new project called Explore2. The Explore2 project was



finally launched in 2021 on the joint initiative of French scientists, the French Ministry of the Environment and the French Biodiversity Agency (OFB), after two years of maturing. The objectives of the project have been defined after collecting feedback from users regarding the results produced by Explore2070 and their use, and based on the availability of data and hydrological models. The two main objectives of the Explore2 project were: (1) updating knowledge on the impact of climate change on hydrology using more recent GHG scenarios and regional climate projections, and (2) better supporting stakeholders

(ministries, water agencies, local authorities, consultancies, economic actors, etc.) in understanding and using data and results to design sustainable water management strategies. The ambition was to go beyond the limitations of the previous project mentioned above (e.g. more simulation points evenly distributed across mainland France, transient simulations spanning the whole of the 21$^{st}$ century…).

**1.2 The Explore2 dataset**

User feedback on the Explore2070 dataset helped define the shape of Explore2. In particular, stakeholders expressed the need for an updated national study that would serve as a reference for hydrology, incorporate new time horizons, with outputs adapted to operational needs, and a larger number of simulation points. Explore2 was structured around two work packages. The first work package was the scientific one. It aimed at providing a benchmark set of climate, river flows, groundwater levels and groundwater recharge projections for mainland France for the 21$^{st}$ century along with a detailed evaluation and support of

the given information. The approach followed a multi-scenario and multi-model approach applied uniformly to assess the uncertainties at the different levels of the climate and hydrological modelling. The second work package focused on user support and exchanges with the stakeholders.

The objectives of this paper are: (i) to describe the climate projections and the hydrological models used to produce the ensemble of simulations in the Explore2 dataset, (ii) to provide an evaluation of the hydrological simulations on the past

climate, and (iii) to present the main results.

Section 2 presents briefly the climate projections produced by Explore2. Section 3 details the rationale behind the multi-model hydrological simulation experiment. Section 4 provides a detailed description of the different hydrological models. Section 5 describes the evaluation of hydrological models. Section 6 details the way cooperation between scientists and stakeholders was organised throughout the project to deliver an actionable Explore2 dataset. Section 7 details the main findings with a

comparison to previous studies. Section 8 concludes.

**2. Climate projections**

This section describes the climate projection dataset used for deriving the Explore2 dataset through the use of hydrological models (HMs). Climate projections were retrieved for different greenhouse gas emission scenarios based on three Representative Concentration Pathways (RCPs) through the use of a hydroclimatic modelling chain composed of global





climate models (GCMs), regional climate models (RCMs), bias correction methods (BCs). The climate data and models that underpin the Explore2 dataset are described in Marson *et al*. (2024), but a brief overview is given below.

## 2.1 Regional model projections

A subset of the EURO-CORDEX ensemble covering Europe (Coppola *et al*., 2021) has been used here as the primary source of climate projections. EURO-CORDEX projections are high-resolution versions of the global projections produced by the

CMIP5 simulation exercise, which served as the basis for the 5[th] IPCC assessment report. The EURO-CORDEX ensemble contains more than a hundred climate projections covering Europe at a 12-km resolution.

## 2.2 Bias correction

To be used as inputs to HMs, RCM outputs were post-processed (including bias correction and downscaling). Two methods BCs were applied in the Explore2 project, namely ADAMONT (Verfaillie *et al*., 2017) and CDF-t (Michelangeli *et al*., 2009).

The two variants of the quantile-mapping approach force the statistical distribution of the simulated variables to match that of the 8-km regular grid SAFRAN atmospheric reanalysis (Vidal *et al.*, 2010) over 1976-2005 that forms the baseline reference climate for the project.

## 2.3 Specific requirements regarding AR6 and uncertainty analysis

Projections from the 5[th] Assessment Report instead from the 6[th] Assessment Report were selected for Explore2 as they were,

at the moment they were retrieved, the only large set of high-resolution projections over France. The climate data have nonetheless been analysed at national level to ensure they convey a message consistent with the latest knowledge of future climate, and we retained only those climate projections that were compatible with the conclusions of the IPCC Working Group I contribution to the 6[th] Assessment Report. This compatibility was assessed by analysing the changes in mean air temperature and precipitation over France in both datasets. A second requirement was the balanced nature of the RCP/GCM/RCM matrix,

needed for a relevant estimation of uncertainty sources in projections (estimation of the contribution from the four sources GCM/RCM/BC/HM to the total uncertainties in future flows), here carried out with the QUALYPSO method (Evin *et al*., 2019; Evin, 2023). Finally, the regional climate projections considered for Explore2 are an ensemble of 72 projections (including 10 with the RCP2.6 low-emission scenario, 9 with the RCP4.5 moderate-emission scenario and 17 with the RCP8.5 high-emission scenario for each BC).

## 2.4 Climate variables

The climate data used for Explore2 are variables required for hydrological modelling: precipitation, air temperature, wind speed, relative humidity, and radiation components. They are provided at a daily time step on square grid cells with an 8-km resolution covering mainland France over the period 1951-2100. Unlike CDF-t, the ADAMONT method was also able to compute and deliver bias-corrected data at the hourly time step. Reference evapotranspiration (ET0) was calculated using the



Penman-Monteith formula (Allen *et al.*, 1998) parameterized for short grass, the net radiation being derived from the Hargreaves equation (Hargreaves and Samani, 1985).

### 2.5 Storylines

Unfortunately, uncertainties are sometimes ignored by stakeholders: For pragmatic reasons (limited computing resources) or because they do not know how to select a subset of projections, only one climate projection or median changes are considered

in prospective studies. To counter these misuse and to responds to user expectation, four storylines were identified from the ensemble of the climate projections. They correspond to four projections under the RCP8.5 high-emission scenario, showing contrasting changes in seasonal precipitation and temperature at the national scale between the period 2070-2099 and the baseline period 1976-2005 (Fig. 1). They do help stakeholders to make informed choices and illustrate climate-related uncertainties at the end of the century. The four storylines were selected from the bias-corrected projections using ADAMONT,

as this BC method is the only one capable of meeting the hourly input data requirements for five hydrological models (Section 4). These are as follows:

  – Green storyline: "Marked annual warming and increased precipitation",

  – Yellow storyline: "Moderate future changes",

  – Orange storyline: "Marked annual warming and drying",

– Purple storyline: "Marked annual warming and high seasonal contrasts in precipitation changes".

Table 1 presents the main characteristics of the four storylines, in terms of annual and seasonal climate changes under the RCP8.5 high-emission scenario at the end of century, together with those of the whole Explore2 climate ensemble.

| Name | Green | Yellow | Orange | Purple | Explore2 |
|---|---|---|---|---|---|
| GCM/RCM/BC modelling chain | MOHC-HadGEM2-ES/ CNRM- ALADIN63/ADAMONT | CNRM-CERFACS-CNRM- CM5/CNRM- ALADIN63/ADAMONT | ICHEC-EC- EARTH/MOHC- HadREM3-GA7- 05/ADAMONT | MOHC-HadGEM2- ES/CLMcom- CCLM4-8- 17/ADAMONT | All |
| **Change in temperature (°C)** | | | | | |
| Year | +4.8 | +3.7 | +4.6 | +5.0 | 3.9 [+2.9 ; +5.2] |
| Winter | +3.8 | +3.2 | +3.7 | +4.2 | 3.4 [+2.7 ; +4.3] |
| Summer | +6.1 | +4.2 | +6.4 | +6.5 | 4.6 [+3.4 ; +7.0] |
| **Change in total precipitation (%)** | | | | | |
| Year | +6 | +6 | -9 | -8 | -1 [-17 ; +10] |
| Winter | +26 | +18 | +12 | +26 | 14 [+3 : +46] |
| Summer | -13 | -10 | -40 | -45 | -25 [-59 ; +5] |
| **Change in reference evapotranspiration (%)** | | | | | |
| Year | +31 | +28 | +43 | +26 | 21 [+10 ; +43] |





Table 1: Characteristics of the four storylines retained for Explore2. Changes are calculated as France-average between the end of
the century 2070-2099 and the baseline period 1976-2005. The median and extreme values (minimum and maximum in brackets)
for the Explore2 dataset for RCP8.5 are shown in the last column.

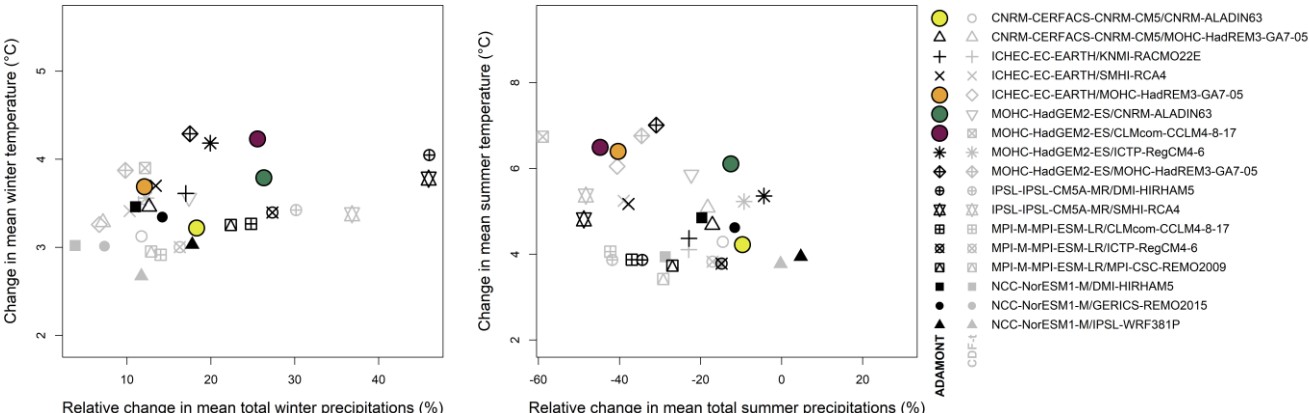

Figure 1: Changes in seasonal precipitation and temperature (winter and summer) between the benchmark period 1976-2005 and
the end of the century 2070-2099 under RCP8.5.

## 3. Hydrological modelling general features

The effect of such climate projections on hydrology was derived through hydrological modelling, and this section describes
the main features of the modelling exercise.

### 3.1 Natural hydrology modelling under climate change

The Explore2 project made use of 12 hydrological models (HMs) used in combination with the ensemble of climate projections
described above. Explore2 HMs cover a wide range of structures and spatial discretization; they include surface and
groundwater hydrological models, and an empirical model used to calculate the potential groundwater recharge. Across this
diversity of models, the Explore2 project provides a "natural reference hydrology" on which to build prospective studies on
water sharing. Streamflows are simulated without any water management action (no water abstraction nor releases, and no
reservoir management), and are therefore considered natural. Computing groundwater levels without including abstractions
was impossible; consequently, these abstractions were kept at their current level in the simulations over the 21st century.
Explore2 HM structures cover a range from conceptual to physically-based structures. Conceptual models – together with the
J2000 model which does not resolve the energy balance – use Penman–Monteith ET0 as presented above, while other models
compute actual evapotranspiration from their water and energy balance models.

Many HMs had to be calibrated. No common calibration rules were adopted by the modellers and calibration was achieved
thanks to the experience of each partner in their model. However, all HMs were evaluated using the same framework, with



common efficiency criteria calculated over a common evaluation period on the same set of reference points. HM model
evaluation is the subject of Section 5.

Most of these models have been extensively used in previous climate change impact and adaptation studies in large French
catchments (see e.g., Amraoui *et al*., 2019; Lemaitre-Basset *et al*., 2024; Morel *et al*., 2022; Seyedhashemi *et al*., 2022; Thirel
*et al*., 2019; Vergnes *et al*., 2023).

| Model | Type | Spatial discretisation | Optimisation of the parameters | Reference | Spatial domain | Number of simulation points | Number of projections | Bias correction method |
|---|---|---|---|---|---|---|---|---|
| CTRIP | Physically-based resolving energy balance | Distributed (regular grid cells) | No | Decharme *et al.* (2019) | France | 2024 | 10 (RCP2.6), 9 (RCP4.5), 17 (RCP8.5) | ADAMONT |
| EROS | Conceptual | Lumped | Automatic | Seyedhashemi *et al.* (2022) | Brittany | 60 | 20 (RCP2.6), 18 (RCP4.5), 34 (RCP8.5) | ADAMONT CDF-t |
| | | Semi-distributed | Automatic | | Loire River basin | 327 | | |
| GRSD | Conceptual | Semi-distributed | Automatic | de Lavenne *et al.* (2019) | France | 3712 | 20 (RCP2.6), 18 (RCP4.5), 34 (RCP8.5) | ADAMONT CDF-t |
| J2000 | Process-oriented | Semi-distributed (hydrological response units) | Manual | Krause *et al.* (2006) | Loire and Rhône River basins | 1291 | 20 (RCP2.6), 18 (RCP4.5), 34 (RCP8.5) | ADAMONT CDF-t |
| MORDOR-SD | Conceptual | Lumped | Automatic | Garavaglia *et al.* (2017) | France | 611 | 20 (RCP2.6), 18 (RCP4.5), 34 (RCP8.5) | ADAMONT CDF-t |
| MORDOR-TS | Conceptual | Semi-distributed | Automatic | Rouhier *et al.* (2017) | Upper Loire River basin | 535 | 20 (RCP2.6), 18 (RCP4.5), 34 (RCP8.5) | ADAMONT CDF-t |
| ORCHIDEE | Physically-based resolving energy balance | Distributed (regular grid cells) | Manual | Boucher *et al.* (2020) | France | 3587 | 10 (RCP2.6), 9 (RCP4.5), 17 (RCP8.5) | ADAMONT |
| SIM2 | Physically-based resolving energy balance | Distributed (regular grid cells) | No | Le Moigne *et al.* (2020) | France | 649 | 10 (RCP2.6), 9 (RCP4.5), 17 (RCP8.5) | ADAMONT |
| SMASH | Conceptual | Semi-distributed | Automatic | Jay-Allemand *et al.* (2020) | France | 3821 | 20 (RCP2.6), 18 (RCP4.5), 34 (RCP8.5) | ADAMONT CDF-t |

**Table2: Characteristics of the nine surface hydrological models and their application within the Explore2 project.**

All surface HMs have produced daily streamflow time series. Some modellers also provided other water-related variables (soil
moisture, snow water equivalent, groundwater storage, etc.), depending on their confidence on such outputs. The Explore2
dataset described we focus on streamflow and groundwater level data, for surface and groundwater HMs, respectively. Table
2 and Table 3 list the main characteristics of the Explore2 HMs, which are described in detail and in alphabetic order, in the
Supplement section S1. Note that, in addition to the input data, certain characteristics are shared by the hydrological models





(e.g. the SURFEX land surface model (Le Moigne *et al.*, 2020) is a common component of the AquiFR, CTRIP and SIM2 models), and that one of the key-features that differentiates the hydrological models for the analyses in the following sections is the choice of parameter optimisation: automatic, manual, or no optimisation.


| Model | Type | Spatial discretisation | Optimisation of the parameters | Reference | Spatial domain | Number of simulation points | Number of projections | Bias correction method |
|---|---|---|---|---|---|---|---|---|
| AquiFR | Physically-based resolving energy balance | Distributed (regular grid cells) | Automatic | Vergnes *et al.* (2020) | Nord-Pas-de-Calais, Basse-Normandie, Bassin parisien, Poitou-Charentes, Alsace, Tarn-et-Garonne | 139,042 grid points 724 piezometers | 10 (RCP2.6), 9 (RCP4.5), 17 (RCP8.5) | ADAMONT |
| EROS | Conceptual | Lumped | Automatic | Seyedhashemi *et al.* (2022) | Brittany | 41 piezometers | 20 (RCP2.6), 18 (RCP4.5), 34 (RCP8.5) | ADAMONT CDF-t |
| MONA | Physically-based | Distributed (regular grid cells) | Manual | Aissat *et al.* (2023) | Aquitaine | 14,388 grid points 416 piezometers | 10 (RCP2.6), 9 (RCP4.5), 17 (RCP8.5) | ADAMONT |
| RECHARGE | Conceptual | Semi-distributed | No | Caballero *et al.* (2021) | France | 621 hydrogeological units | 10 (RCP2.6), 9 (RCP4.5), 17 (RCP8.5) | ADAMONT |

**Table3: Characteristics of the groundwater hydrological models and their application within the Explore2 project.**

### 3.2 Forcing data

Models requiring hourly input data (AquiFR, CTRIP, ORCHIDEE, and SIM2) could not be forced with projections bias-corrected at the daily time step by the CDF-t method. They were therefore forced only by the 36 projections bias-corrected
with ADAMONT. Due to a lack of resources, the two hydrological models RECHARGE and MONA have been forced by climate projections using ADAMONT only.

### 3.3 Transient simulation outputs

Explore2 HMs provided transient daily simulations of streamflow and groundwater, for surface and groundwater HMs, respectively, over the period 1976-2005, as forced by GCM/RCM/BC projections described above. The period 1950-1975 was
used as a warm-up period.

### 3.4 Streamflow simulation network

Simulation points were selected based on several operational and thematic national networks. These points are de facto of interest for different management issues (knowledge, monitoring, regulation, etc.) and aim at providing a uniform national coverage. The national networks considered for the selection are:





200       –      The network of gauging stations in the Hydroportail database (http://hydro.eaufrance.fr).

      –      The Observatoire National Des Étiages (ONDE, https://onde.eaufrance.fr), a monitoring network of qualitative knowledge about summer low flows and streamflow intermittence.

      –      The network of monitoring points of the Schémas Directeurs d'Aménagement et de Gestion des Eaux (SDAGE) and of the Réseau de Contrôle de Surveillance (RCS) used for the implementation of the European Union's Water

205                  Framework Directive (2000/60/EC, WFD) and the achievement of "good" status of water bodies.

Combining the different networks resulted in more than 10,000 candidate points. Given the 8-km spatial resolution of the gridded climate projections that were used as forcings for hydrological modelling, points with a drainage area of less than 64 km² have been excluded. A procedure was carried out to retain only one point when two monitoring points were too close, based on comparing the distance between points to a threshold. Finally, a set of 4,043 locations of interest was identified.

**3.5 Surface hydrographic network and actual simulated points**

Surface hydrological models need to develop a river network to describe the upstream-downstream link between catchments. Each surface HM has its own river network derived from Digital Terrain Models (DTMs). Any HM river network may deviate from the actual location of the watercourses. Simulation points must be positioned on the river network to provide hydrological projections. Decisions were made to move simulation points on the modelled river network with respect to the actual drainage

area. For example, for GRSD, SMASH and ORCHIDEE, the allowed point shift was limited to 2 km from the actual location for GRSD and SMASH, and 5 km for ORCHIDEE, with a relative error of less than 20 % between the actual drainage area and that drainage area derived from the modelled river network. In sectors known to be karstic, these constraints were relaxed. If it was impossible to find a location that both respects the distance and surface constraints, this simulation point was discarded from the HM network and therefore no hydrological projection was produced for this point. This is why HMs with national

coverage do not provide projections at all simulation points identified above. Finally, hydrological projections are available at ~4,000 simulation points (Fig. 2c) and there are projections from at least four HMs for ~2,400 locations. Some regions that do not have an extensive river network may have only a few simulation points (e.g. Atlantic coast of south-west France). Due to the regional coverage of J2000, EROS and MORDOR-TS, the Loire River basin is the best documented catchment with projections available for all HMs (9) for 31 simulation points. The MORDOR-SD model has provided projections only for the

reference points. Figure A1 (Supplement, S1) shows the location of simulation points for each surface HM and complements the overall output network shown in Fig. 2c. The SIM2 model does not provide simulations for catchments smaller than 173 km². It is noteworthy that this model exhibits the coarsest resolution (median size: 984 km² for SIM2 against ~240 km² for the other models). Finally, the median size of the catchment area associated with the simulation points is 240 km² for the Explore2 project while the median size of the catchment with hydrological simulations in Explore2070 was 640 km². The Explore2

project provides almost three times as many simulation points as Explore2070 with a finer spatial resolution.





(a)

(b)

(c)

**Figure 2: Partition of water bodies (MESO) used for potential recharge of groundwater (a), and sets of simulation grid cells and points where water level (b) and streamflow (c) data are simulated.**



### 3.6 Groundwater simulation grid

The potential recharge of groundwater by precipitation is estimated at the regional scale. The pre-existing division of France into 621 groundwater bodies (Masses d'Eau SOuterraine, MESO) delineated so as to be homogeneous in terms of hydrogeological characteristics (Mardhel and Normand, 2006) has been adopted here. In addition to the physical justification of their use, the partition of MESO units is also considered for operational issues, i.e. for assessing the quantitative and qualitative groundwater status under the Water Framework Directive (WFD) in France. As part of the previous Explore2070 project, groundwater simulations were provided for a group of separated but major aquifers with sedimentary formations (excluding karst), which are also the most heavily exploited. The domain simulated by the groundwater HMs was extended in the Explore2 project to cover a large portion of north-western mainland France. Groundwater levels are provided on a 1-km resolution regular grid for the AquiFR modelling platform (~140,000 points), and on a 2-km resolution regular grid for MONA (~10,000 points). In addition to gridded outputs, simulations are provided on a set of piezometers (~1,100 points), all of which are listed in the national database ADES (https://ades.eaufrance.fr/) and Hub'Eau (https://hubeau.eaufrance.fr/). The hydrogeological units used for estimating the potential recharge of groundwater, and the grid cells and the simulation points of the groundwater HMs are displayed in Figs. 2a and 2b, respectively.

### 4. Evaluation of the modelling chain

### 4.1 Method

The evaluation process was conducted in two phases. First, Explore2 HMs have been evaluated under past climate conditions by comparing simulations forced by the meteorological reanalysis SAFRAN (Vidal *et al*., 2010) with the observations at the reference points from the networks described above. The chosen period (1976-2019) for evaluation covers the baseline reference climate period (1976-2005) and extends up to year 2019, which was the last year of complete observations at the beginning of the project. The evaluation is a prerequisite before studying the impact of climate change on water resources, i.e. in conditions where all HMs are applied in extrapolation (Thirel *et al*., 2015). The ability of each HM to simulate different hydrological characteristics at a given simulation point is therefore essential. Over and above operational needs specific to the Explore2 project, the performance assessment carried out allows identifying processes that are poorly simulated by HMs and suggesting ways of improving them for further applications. Note that there are no direct observations of potential groundwater recharge and the evaluation of RECHARGE is extensively detailed in Robelin *et al*. (*in prep.*).Second, the entire modelling chain from GCMs to HMs was examined to assess how well the HMs forced by historical runs can reproduce the various facets of the hydrological regime. Here, the focus is on surface hydrological models.



### 4.1.1 The reference network for evaluation

Hydrological time series available in the national databases Hydroportail (https://hydro.eaufrance.fr) and ADES (https://ades.eaufrance.fr), as well as in an internal database of one project partner (EDF), were used as benchmark data for the evaluation. Not all monitoring sites referenced in these two databases are relevant for the performance assessment depending on the level of human-induced alteration on natural functioning of the rivers or aquifers, the data availability, the quality of the measurements, and the size of the catchment (relevant for the selection of gauging stations). The level of alteration is partly assessed by comparing observed flows (mean annual and low-flows) to estimated consumptions derived from declared water withdrawals (https://bnpe.eaufrance.fr), and to the total volume of reservoir capacities over the upstream catchment. The final set of reference gauging stations considered to be relevant for model evaluation has been identified through a collective expertise (Strohmenger *et al*., 2023) and has been validated by the members of two user committees (see more on these user committees in the next section). These reference points are representative of near-natural conditions in accordance with the simulation assumptions. For the evaluation of the surface HMs, 611 gauging stations have been identified. The lengths of the time series range between 23.5 and 44 years, with a median of 40.6 years (Strohmenger *et al*., 2023). For the evaluation of the groundwater HMs, 227 reference piezometers have been selected, and also validated by the user committees. They have at least 10 years of data, 97 % have 20 years of data and 71 % have 27 years of data.

### 4.1.2 Performance metrics

The performance assessment involves a set of metrics which evaluate diverse aspects of streamflow and groundwater regime at several time scales, from daily to long-term.

A set of 12 metrics and two tests were selected to evaluate the surface HMs across the flow regime. These metrics, detailed in Table 3, are complementary and evaluate different aspects of the hydrological time series: overall performance, climate sensitivity, high flows, mean flows, low flows, and robustness. Criteria assessing the overall performance are the Bias and the KGE√ (giving roughly equivalent weight to high flows and low flows). The sensitivity of streamflow to precipitation and temperature is assessed through climatic elasticity indices ($\varepsilon_{R, DJF}$, $\varepsilon_{R, JJA}$, $\varepsilon_{T, DJF}$, $\varepsilon_{T, JJA}$, Sankarasubramanian *et al*., 1991), which may help understanding contrasting responses of HMs under climate change. High, mean, and low flows and associated processes are assessed with a set of 6 hydrological signatures (Q10, *medt*QJXA, aCDC, αQA, Q90, *medt*VCN10). The choice of the metrics was partly supported by the analysis of the correlation matrix between pairs of around 50 metrics suggested by modellers (highly correlated metrics were removed from the initial set of 50 metrics). Finally, the robustness assessment test (RAT, Nicolle *et al*., 2021) is based on the identification of a correlation between model errors and temperature (RATT) or precipitation (RATR). The Spearman correlation ρ between model bias and meteorological variables was computed, and the robustness for the tested HM is rejected when a significant correlation is found (p-value = 10%). Four metrics were considered for the evaluation of groundwater HMs and are detailed in Table 4: $NSE_{bias}$, $NSE_{SGI}$, r, and Bias. Three of them were included



in the previous assessment of groundwater HMS by Vergnes *et al.* (2020). For each criterion described above, a range defining an acceptable fit was proposed by modellers and shown in Table 3 and Table 4.

| Metrics | Category | Definition | Value for a perfect fit [acceptable fit] |
|---|---|---|---|
| **KGE√** | Overall performance | Kling Gupta efficiency (Gupta *et al.*, 2009) on square-root daily streamflow | 1 [0.5; 1] |
| **Bias** | | Relative bias | 0 [-0.2; 0.2] |
| $\varepsilon_{R, DJF}$ | Climate sensitivity | Rainfall elasticity in winter DJF and in summer JJA (ratio) | 1 [0.5; 2] |
| $\varepsilon_{R, JJA}$ | | | |
| $\varepsilon_{T, DJF}$ | | Temperature elasticity in winter DJF and in summer JJA (ratio) | |
| $\varepsilon_{T, JJA}$ | | | |
| **Q10** | High flows | Relative error in flow that is exceeded 10 % of the time | 0 [-0.2; 0.2] |
| *med***tQJXA** | | Bias in median occurrence (Julian day) of the annual maximum daily streamflow | 0 [- 30 days; 30 days] |
| **aCDC** | Mean flows | Difference between flow that is exceeded 66 % of the time and flow that is exceeded 33 % of the time (ratio) | 1 [0.5; 2] |
| **αQA** | | Sen slopes computed on annual flows (ratio) | 1 [0.5; 2] |
| **Q90** | Low flows | Relative error in flow that is exceeded 90 % of the time | 0 [-0.8; 0.8] |
| *med***tVCN10** | | Bias in median occurrence (Julian day) of the annual minimum of the 10-day mean flow | 0 [- 30 days; 30 days] |
| **RAT$_T$** | Robustness | Robustness assessment test (Nicolle *et al.*, 2021) against annual air temperature | Passed at the 10 % confidence level |
| **RAT$_R$** | | Robustness assessment test (Nicolle *et al.*, 2021) against annual precipitation | Passed at the 10 % confidence level |

**Table 3: Metrics used for the evaluation of streamflow simulations.**



| Metrics | Category | Definition | Value for a perfect fit [acceptable fit] |
|---------|----------|------------|------------------------------------------|
| **NSE**$_{bias}$ | Overall performance | Nash-Sutcliffe efficiency coefficient on deviation of groundwater level to the mean | 1 [0.5; 1] |
| **NSE**$_{SGI}$ | | Nash-Sutcliffe efficiency coefficient on Standardised Groundwater level Index | 1 [0.5; 1] |
| **r** | | Correlation coefficient | 1 [0.5; 1] |
| **Bias** | | Mean error | 0 [-2 m; 2 m] |

**Table 4: Metrics used for the evaluation of groundwater level simulations.**

## 4.2 Evaluation results

This section provides an overview of the evaluation of the HMs over the 1976-2019 period. Note that the models had their parameters obtained after comparing the simulations against observed data available over the period 1976-2019 and therefore the evaluation as described below is not systematically an evaluation with independent data. More details are available in (Sauquet *et al.*, 2023).

### 4.2.1 Evaluation of surface hydrological models

Surface HMs show a good overall performance when forced by the SAFRAN meteorological surface reanalysis (see the distribution of four selected scores in Table 5). They perform well across the whole streamflow range: the majority of the evaluation scores fall within the range considered acceptable by the modellers (for more than 66 % of the simulations, all models and reference stations combined).The performance of all models is good to very good for large catchments (only 3 of the 25 catchments with surface area greater than 5,000 km² have KGE√ < 0.5), whereas it is more scattered for small rivers

(254 catchments with surface area less than 200 km² with KGE√ between -4.4 and 0.98, and 120 of them have KGE√ < 0.5). The acceptable fit (Table 3) in terms of KGE√ and Bias are met by all surface HMs for 183 of the 611 reference gauging stations. Conversely, for 63 of the reference stations, the evaluation scores are mostly (at least for half of the HMs) outside one of the two acceptable ranges for KGE or Bias. Note that the global performance of the hydrological models CTRIP, J2000, and ORCHIDEE varies widely between reference stations. The variation in global performance should encourage data users

to examine the relevance of these models locally and subsequently their interest in using the hydrological projections produced by these HMs.

Models perform well for high flows, with a median performance across the set of reference stations being close to zero for the Q10 metric. The performance is globally lower for low flows with possible biases for two HMs: SIM2 with a median score Q90 of 1.03 (103 %) tends to overestimate low flows for most catchments while CTRIP with a median score Q90 of -0.43 (-

43 %) tends to underestimate low flows for most catchments.





Unsurprisingly, the evaluation scores of models whose parameters are obtained by automatic numerical optimisation (GRSD, EROS, MORDOR-TS, MORDOR-SD, and SMASH) over the period 1976-2019 are close to the perfect fit (Table 3). For example, the GRSD model was calibrated against streamflow using the KGE√ objective function, leading obviously to very high values for this metric.


| HM | KGE√ | Bias | Q10 | Q90 |
|---|---|---|---|---|
| CTRIP | 0.63 [-1.73; 0.94] | 0 [-0.9; 5.72] | 0.04 [-0.91; 8.27] | -0.43 [-1; 17.47] |
| EROS | 0.89 [0.22; 0.96] | -0.03 [-0.22; 0.17] | -0.06 [-0.24; 0.15] | 0.21 [-0.33; 3.19] |
| GRSD | 0.92 [0.33; 0.97] | 0 [-0.27; 0.96] | -0.01 [-0.37; 1.04] | -0.04 [-0.74; 3.33] |
| J2000 | 0.78 [-2.72; 0.93] | 0.01 [-0.55; 7.83] | 0 [-0.54; 9.39] | 0.23 [-0.83; 12.2] |
| MORDOR-SD | 0.94 [0.58; 0.98] | 0 [-0.1; 0.23] | 0.01 [-0.18; 0.29] | -0.1 [-0.83; 2.33] |
| MORDOR-TS | 0.9 [0.71; 0.97] | 0.01 [-0.23; 0.31] | 0 [-0.2; 0.62] | -0.01 [-0.58; 2.23] |
| ORCHIDEE | 0.65 [-4.37; 0.89] | 0.06 [-0.97; 8.28] | 0.09 [-0.96; 9.28] | -0.28 [-1; 5.05] |
| SIM2 | 0.72 [-0.6; 0.94] | 0.06 [-0.88; 0.91] | 0 [-0.78; 1.29] | 1.03 [-1; 15.93] |
| SMASH | 0.89 [0.3; 0.96] | 0.01 [-0.2; 0.21] | 0.04 [-0.27; 0.61] | -0.05 [-0.97; 4.75] |

**Table 5: Evaluation metrics of surface HMs (median [minimum; maximum]) forced by SAFRAN for the 1976-2019 period over all available reference gauging stations combined. Note that some models were not applied over all 611 reference gauging stations.**

The proportion of values within the acceptable range for all models and all stations is displayed in Fig. 3 for the 14 evaluation
scores. Less satisfactory results in terms of KGE√ are found in the north of France, the downstream tributaries of the Seine River basin, and the South-East. These poor results are primarily due to unsatisfactory modelling of surface-groundwater exchanges for these areas. Surface HMs encounter difficulties for groundwater-dominated catchments located in the North, and are challenged by extended karst systems in the South-East. In the Alps, the low values of KGE√ could be due to difficulties in modelling snow dynamics and biases in the SAFRAN forcings, i.e. an underestimation of total precipitation at high altitude
(Magand et al., 2018). Care must also be taken not to over-interpret KGE√ scores on rivers influenced by groundwater, as this score is largely driven by the variability of streamflow, which is limited for these rivers. The map derived from the Bias score shows that the lowest performing catchments are located to the south-east and north, while the highest performance is found in the west. The analysis of the elasticities $\epsilon$ shows that the response to rainfall pulses in winter ($\epsilon_{P, DJF}$) is well captured by the models. In a way, this result was expected: the models were designed to reproduce the rainfall-flow transformation. There is
good sensitivity ($\epsilon_{T, JJA}$) to air temperature in summer, especially in the south of France where evapotranspiration is the main driver of flow variability. For the other seasons, results are diverse, and in summer, we may suspect a difficulty in reacting well to local precipitation captured only in part by the SAFRAN reanalysis. The average seasonality of floods ($med$tQJXA) is correctly reproduced over France except for southern France. One possible reason is the bimodality of flood seasonality in



these areas (see e.g. Fig. 3 in Blöschl *et al.*, 2017) which calls into question the relevance of computing an average Julian date

of floods. The RAT metrics are very demanding, with a high failure rate, whether the models are calibrated through

optimization or not. Performance in low flows is less satisfactory than in high flows when comparing Q90 and Q10 values.

The acceptable range of variation for Q90 has been chosen wider than for Q10 (Table 3), so that the map associated with Q90

does not lead to widespread failure.



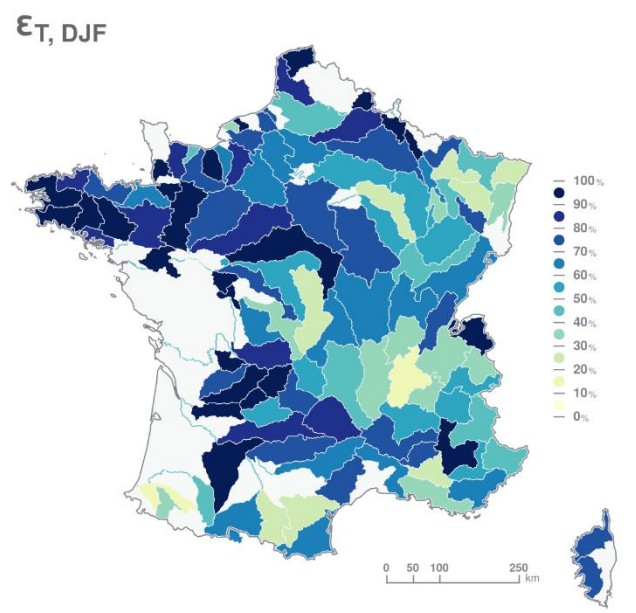

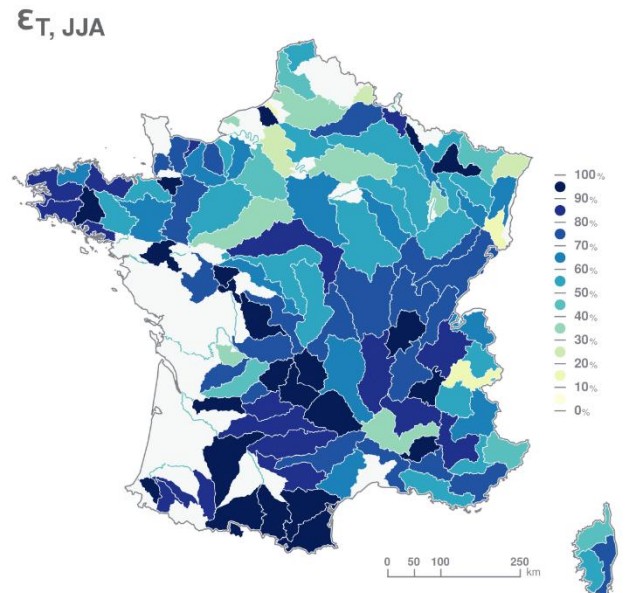

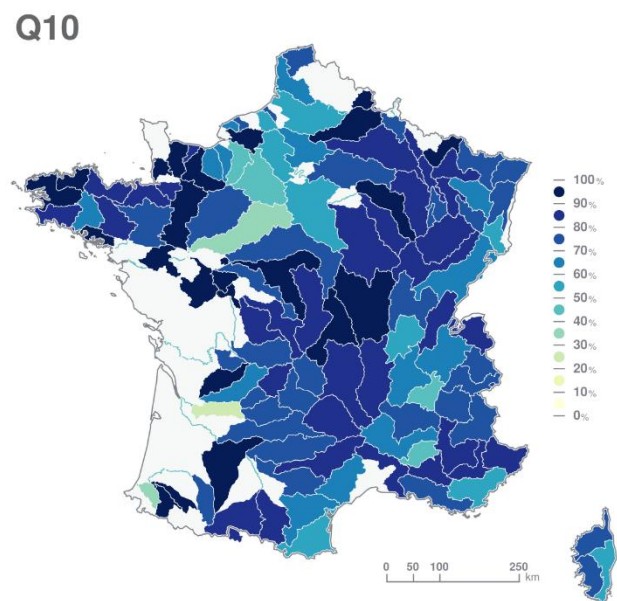

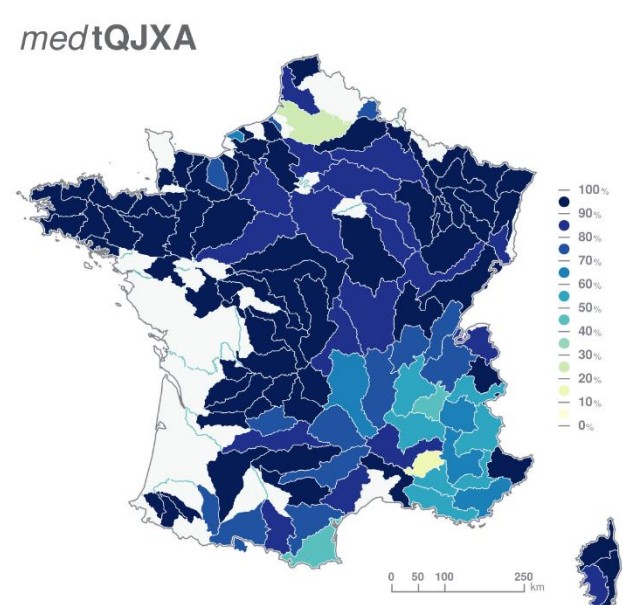







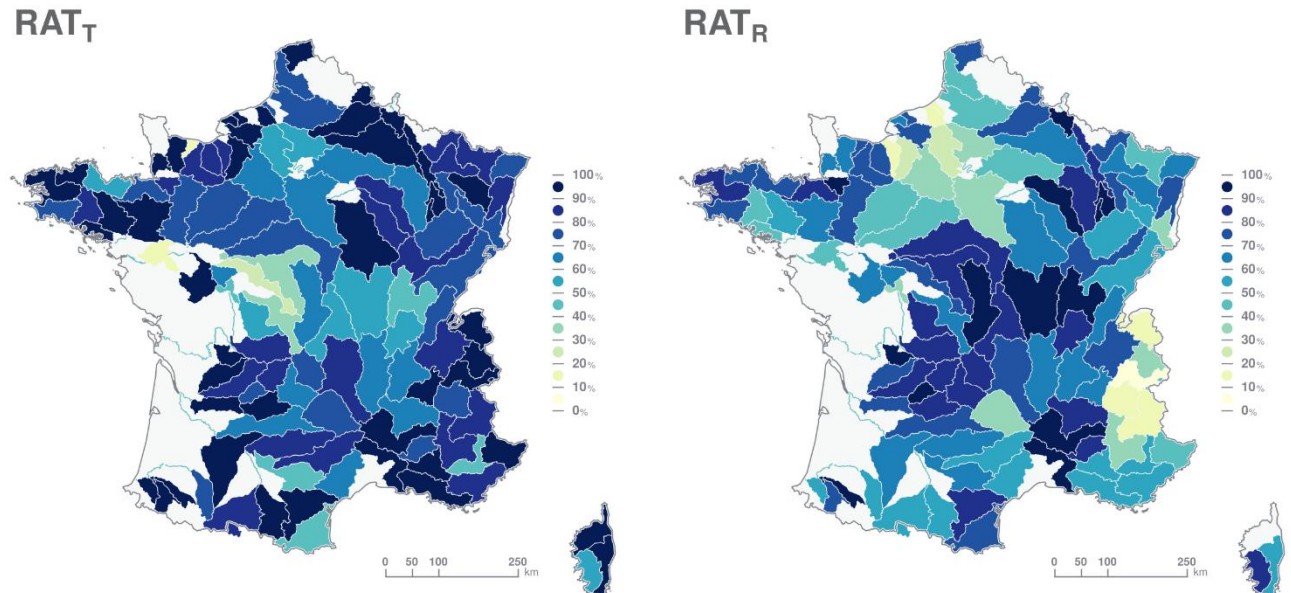

**Figure 3: Proportion (%) of simulations with evaluation metrics within the acceptable range. A value of 100 % means that all HMs demonstrate numerical metrics within the acceptable range at all reference gauging stations while a null value means that all HMs demonstrate numerical metrics outside the acceptable range at all reference gauging stations. Statistics of the evaluation metrics have been computed on a division of France into 187 sub-basins (https://www.sandre.eaufrance.fr) to identify contrasts in HM performance over the 611 reference gauging stations.**





## 4.2.2 Evaluation of groundwater hydrological models

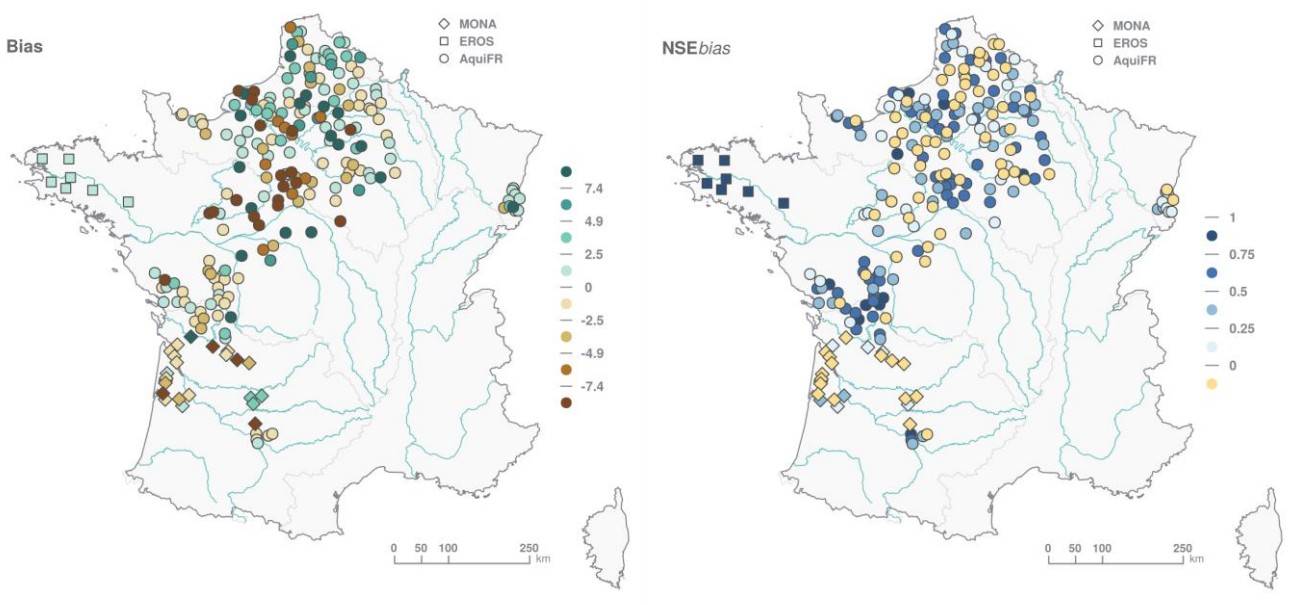


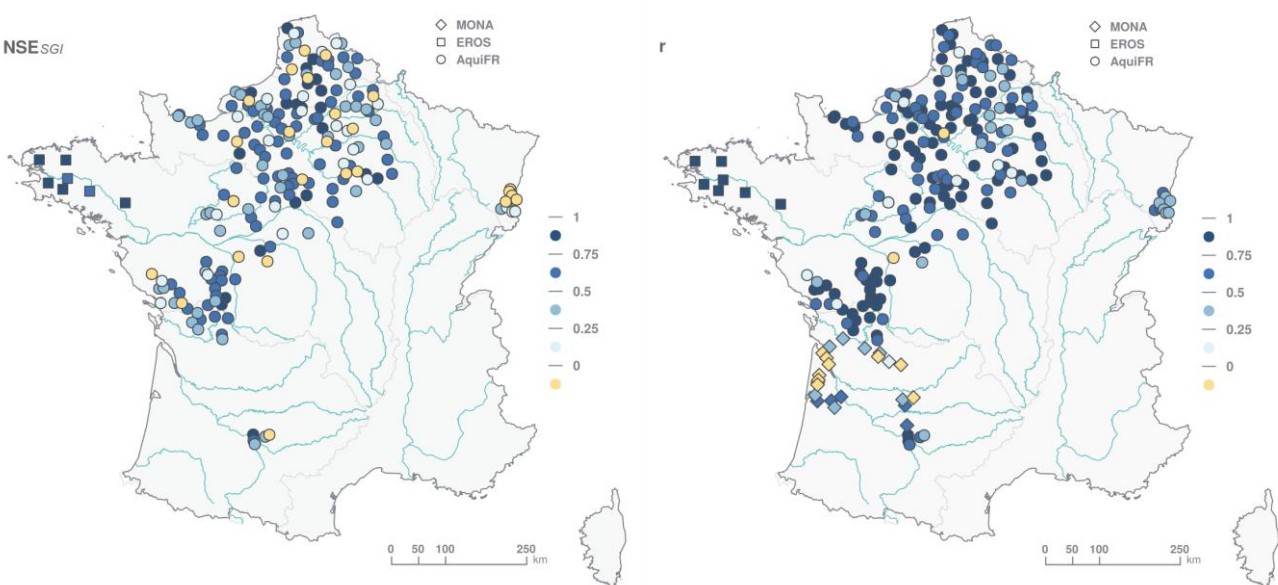

**Figure 4: Evaluation scores obtained for the three groundwater HMs at the reference piezometers.**



Out of the three groundwater HMs, the EROS model is the only one to be calibrated through a numerical optimisation on both observed streamflow and groundwater level observations, leading to very good assessment scores and low biases (Fig. 4). The

evaluation scores obtained with MONA and AquiFR reveal a more heterogeneous match between observations and outputs (Fig. 4).

For the north of France, and more precisely the area covered by AquiFR, the evaluation confirms results recently obtained by Vergnes *et al*. (2020). The AquiFR model shows no systematic bias, with negative bias in the north of the Loire basin, corresponding to the Beauce region, and with positive bias values in the Alsace plain. Elsewhere, there is no significant pattern.

The $NSE_{SGI}$ score is mostly positive (75 % of the values are above 0.25), demonstrating the ability of AquiFR to capture monthly anomalies. The correlation coefficient $r$ is negative on only two of the 197 reference piezometers modelled by the AquiFR platform. In Brittany, all seven reference piezometers simulated by EROS present a correlation coefficient $r$ higher than 0.93, as a consequence of the calibration process.

Results are less satisfactory in the south-west with MONA ($r < 0$ for 10 of the 23 reference piezometers). It should be noted

that MONA is an annual model dedicated to deep aquifers where groundwater levels vary little from one year to the next. Its main objective is to reproduce long-term average groundwater levels in order to study their sensitivity to climate change. The $NSE_{SGI}$ metric was therefore not computed, reducing the number of dots in Fig. 4 for this metric.

### 4.2.3 Evaluation of surface hydrological models driven with baseline climate projections

Assessing the entire chain fed by historical runs is not straightforward. GCM-RCM runs, even bias-corrected ones, are not

expected to reproduce the sequencing of meteorological events observed over the historical period, which makes a criterion such as KGE√ irrelevant for the purposes of the diagnosis. A comparison between simulated and observed hydrological signatures is more appropriate. However, it remains challenging to distinguish between the differences attributable to modelling biases and those due to climate variability, based on this comparison.

Following on from section 5.2.1, the analysis focuses here on surface HMs and on three metrics encompassing the different

phases of the river flow regime: the two percentiles $q90$ and $q10$, which are exceeded 90 % and 10 % of the time respectively, and the mean annual streamflow $qa$ at the 611 gauging stations. These three metrics $X$ have been computed over the common baseline period 1976-2005 from observed streamflow time series ($X_{obs}$), simulated streamflow time series obtained with HMs forced by the reanalysis SAFRAN ($X_{rea}$) and simulated streamflow time series obtained with HMs using historical runs ($X_{hist}$). A global dimensionless criterion ($R_{RH}$) for each metric $X$, each historical run, and each gauging station $i$ of the reference

network is computed as follows:

$$R_{RH}= (X_{rea} - X_{hist})/(X_{rea} - X_{obs}) \tag{1}$$

The value ($X_{rea} - X_{hist}$) is the difference between the two estimates $X_{rea}$ and $X_{hist}$, resulting from combined bias of the climate projections and the effect of the internal variability of the climate. The value ($X_{rea} - X_{obs}$) is the deviation to the observed value $X_{obs}$, resulting from bias in hydrological modelling and in the meteorological reanalysis. Values between -1 and +1 are expected

for *RRH* (i.e. significant additional biases do not appear when HMs are fed by historical runs than when HMs are fed with the





reanalysis). The results are presented in Fig. 5, with boxplots summarising results across the reference network for each metric and for each HM. Here we consider the 17 GCM- RCM runs that have been corrected using the ADAMONT method, as well as the 17 GCM-RCM runs that have been corrected using the CDF-t method.

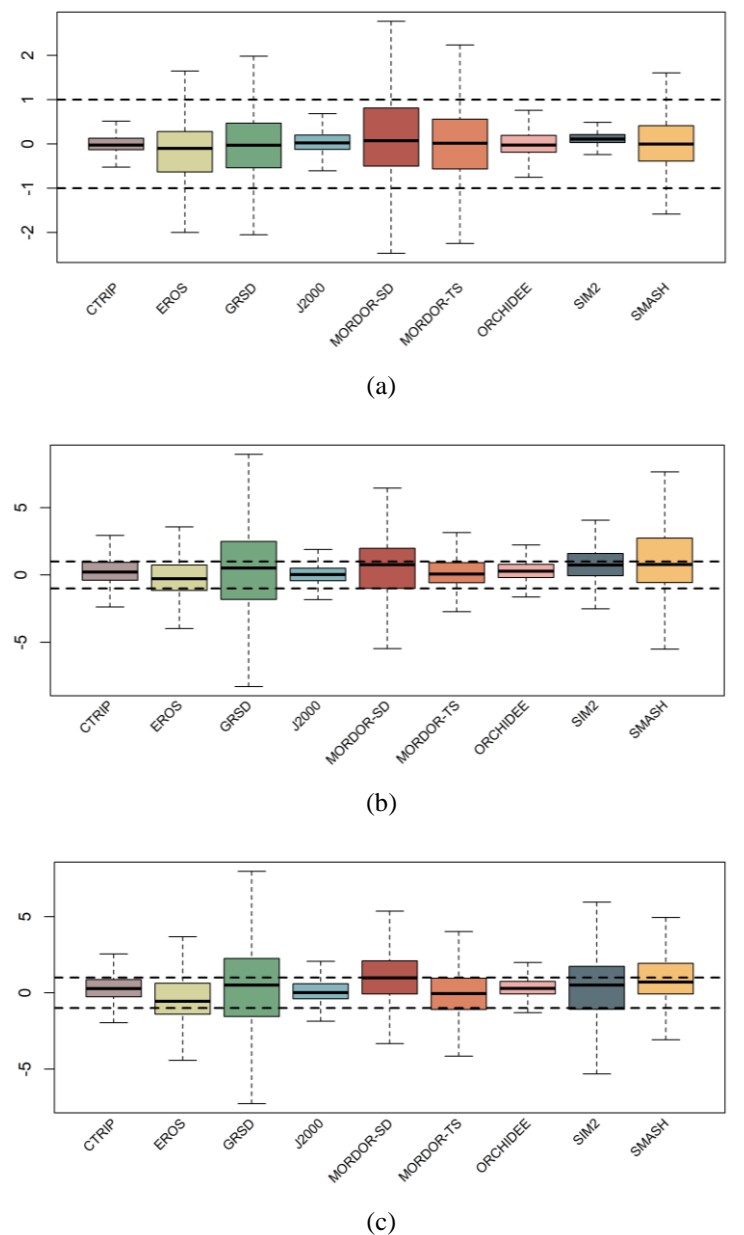

**Figure 5: Boxplots of the RRH criterion computed for the flow quantile that is exceeded 90 % of the time (a), the mean annual streamflow (b), and the flow quantile that is exceeded 10 % of the time (c). The bottom and top of the box are the 25th and 75th percentiles, and the line is always the median. The dashed lines represent the values 1 and -1. The boxes have been ordered from left to right in alphabetical order by model name.**






Figure 5 shows that whatever the variable, the climate modelling chain and the hydrological model, the medians of *RRH* values fall within the -1 to 1 range. However, the medians may deviate from the null value (expected value in the absence of both internal variability and of bias in climate modelling) for both $q10$ and $qa$. For these two metrics, ~45 % and ~65 % of the *RRH* values fall within the [-1; 1] and [-2; 2] intervals, all HMs combined. Globally, there is a high degree of similarity between boxplots obtained with $q10$ and $qa$, but a clear distinction is evident between them and the boxplots obtained with $q90$. The median of *RRH* obtained with $q90$ are very close to zero and ~75 % and ~85 % of the *RRH* values all HMs combined are found in the [-1; 1] and [-2; 2] interval, respectively. The inaccuracies in the representation of low flows by HMs appear to be of a greater order of magnitude than those caused by imperfections in climate modelling. Conversely, the inherent limitations of hydrological and climatic models result in biases of a similar magnitude for mean and high flows. There is a difference in *RRH* spread between the models - particularly for $q90$: the dispersion in *RRH* values is greater for HMs with numerically optimised parameters (EROS, GRSD, MORDOR-SD, MORDOR-TS and SMASH) (i.e. explicit strategies in the calibration process search for $(X_{rea} - X_{obs})=0$ and even small values of $(X_{rea} - X_{hist})$ may lead to high values of *RHH*) while the other models (CTRIP, J2000, ORCHIDEE and SIM2) do not perform as well with possible large deviation to observed streamflow data (i.e high values for $(X_{rea} - X_{obs})$) leading more likely to low *RRH* values. Finally, note that *RRH* values can be negative, indicating possible bias offsetting effects (i.e. $(X_{rea} - X_{hist})$ and $(X_{rea} - X_{obs})$ may display an opposite sign). The use of bias corrected projection from the climate modelling chain introduces naturally additional biases. However, theses biases are limited and river flow seem reasonably approximated by HMs fed by historical runs.

## 5. Co-construction between scientists and stakeholders

The way of presenting these results of the Explore2 project is the results of a close cooperation between scientists and stakeholders from various backgrounds but with interest in water management. This co-construction was organised as part of the user support work package, which was based on exchanges with committees, requested throughout the project. These exchanges notably focused on (1) data requirements (i.e. the most relevant variables and indices for water management among the outputs of the hydrometeorological modelling chains), (2) the structure and topics of a massive open online course (https://e-learning.oieau.fr/enrol/index.php?id=3799) dedicated to the project, (3) the way results were presented in graphs, (4) the contents of technical guides for understanding complex concepts such as uncertainty in climate change projections, and (5) the contents of prototype fact sheets summarising the changes projected at each simulation point. Through this work package, scientists were expected to have a better understanding of how to adapt their outputs to the needs of stakeholders, who in turn could acquire good practice in using the data produced. Both aimed at co-constructing information that is useful for adaptation, and finding scientifically relevant ways of representing and summarising the results obtained by the simulations. These committees have also enabled scientists to obtain local information for building the reference and simulation networks. As an example of cooperation between scientists and stakeholders, the results of present-day evaluation (sections 5.2.1 and 5.2.2) were summarised in the form of summary sheets displaying scores for each HM from local (Supplement S3, Fig. S2 at one



reference gauging station) to regional (Supplement S3, Fig. S3 on the scale of the national hydrogeological units) scales. These
       sheets are publicly available through the open platform for French public data dedicated to the Explore2 project
       (https://entrepot.recherche.data.gouv.fr/dataverse/explore2). This analysis enables users to make a fully informed choice, e.g.
       by eliminating models that do not meet their requirements in terms of summer flows or floods when different surface HMs (1
       up to 9) have provided hydrological projections at a simulation point of interest. The evaluation helps users to possibly make

a pre-selection of surface HMs to be used in prospective studies on water sharing. Groundwater projections were obtained by
       running HMs over separate areas, so there is no possible HM pre-selection there and, finally, the reference piezometers have
       not been the subject of summary sheets.

## 6. Summary of the projected changes at the end of the century

       Results obtained with the AquiFR modelling platform on groundwater levels, on river flow intermittence and on floods are

presented in Jeantet *et al*. (2025), Jaouen *et al.* (2024) and in Tramblay *et al.* (2025).

       This section summarizes the main findings on change in hydrological metrics characterising water resources: the seasonal
       flows (*QDJF*, *QMAM*, *QJJA*, *QSON*), the annual flows *QA*, and the annual recharge with the three GHG emission scenarios
       RCP2.6, RCP4.5 and RCP8.5. Figures 6 and 7 show the maps of the projected median changes between the reference period
       1976-2005 and the end of the century 2070-2099. Dispersion of the median changes across France is summarised by the

interquartile range *IQR* of the median changes between the reference period 1976-2005 and the two time slices 2041-2070
       ('mid-century') and 2070-2099 (Table 7).

       A multi-model index of agreement (*MIA*, Tramblay and Somot (2018)) is also computed on time series to measure the degree
       of agreement between models at each simulation point (Fig. 8):

$$MIA = \frac{1}{N}\sum_{k=1}^{N} i_k \tag{2}$$

where *N* is the total number of projections, $i_k = +1$ when the projection *k* suggests an increase in the considered variable,
       $i_k = -1$ when the projection *k* suggests a decrease in the considered variable, and $i_k = 0$ otherwise. *MIA* values vary between
       -1 and +1, taking a value close to 1 and to -1 when the majority of the *N* projections suggest an increase and a decrease,
       respectively. Trends over the 1976-2100 period were assessed using the non-parametric Theil-Sen slope estimator (Theil,
       1950; Sen, 1968) regardless of the significance of the trends. In accordance with the evaluation of the robustness in the IPCC

WGI Interactive Atlas, the multi-model agreement on the projected changes is considered strong when $|MIA| \geq 0.6$ (more than
       80 % of the projections agree on the sign of the change). We consider also a moderate agreement when $0.6 > |MIA| \geq 0.4$. The
       maps of the multi-model agreement index *MIA* by RCP allows to identify areas of sign agreement. Unfortunately, the number





of projections of annual recharge was considered too small (e.g. only nine for RCP4.5) to allow calculation of related *MIA* values.

Although the annual recharge and annual streamflow have been estimated independently (with recharge not being included in the surface hydrological models), the projections are very consistent with each other in terms of intensity and spatial pattern of median changes (Fig. 6). Most of the regions demonstrate positive changes in both metrics for RCP2.6 while a gradient north-south is suggested for RCP4.5 and RCP8.5 with projected positive changes in the northern France and negative changes in the southern France. Most of the changes in *QA* between the reference period and the two time slices are limited (*IQR* ⊂ [-

10; 10] %) (Table 6). Finally, there is no agreement between projections (|*MIA*| ≥ 0.6 for less than ten simulation points) under RCP2.6 and RCP4.5, while hydrological projections do agree for 12 % of the simulation points across France and these points are located in the southern France and Corsica pointing towards a decrease in *QA* under RCP8.5 (Fig. 8). There is no agreement for an increase in mean annual flows under RCP8.5.

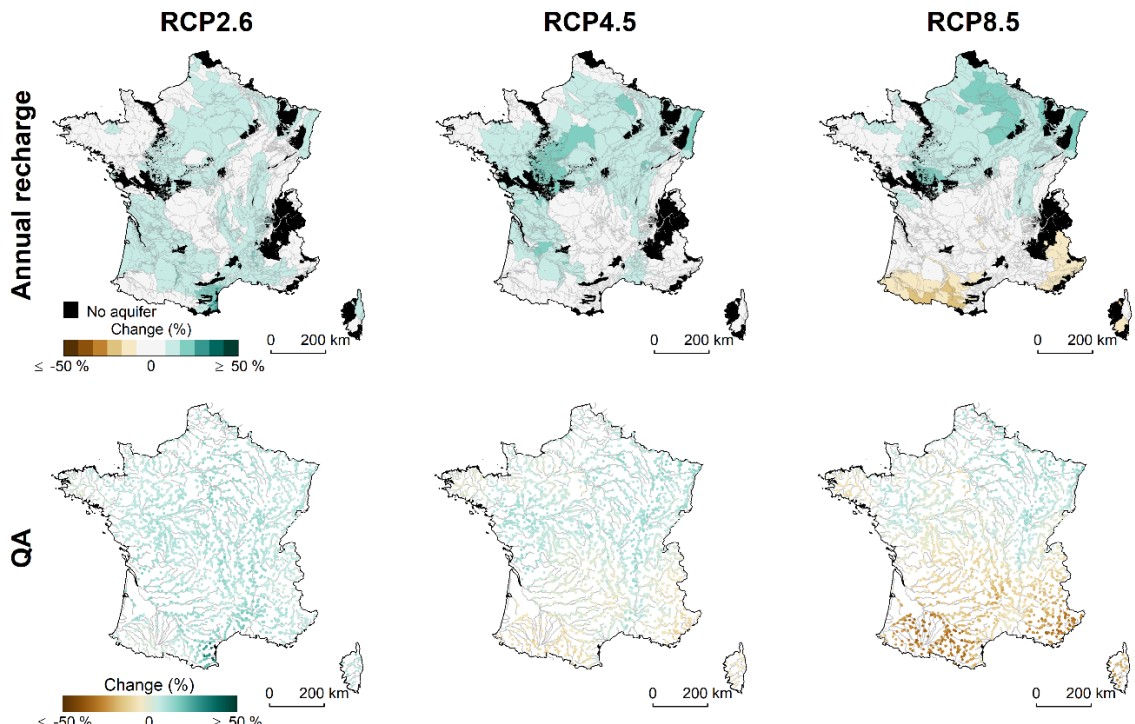

**Figure 6: Median changes in the annual recharge and the annual flow QA between the benchmark period and the end of the century for the three RCPs. The areas coloured black on the maps are those where there is no aquifer and where the notion of recharge is not relevant.**



**Figure 7: Median changes in the seasonal flows QDJF, QMAM, QJJA, and QSON between the benchmark period and the end of the century for the three RCPs.**






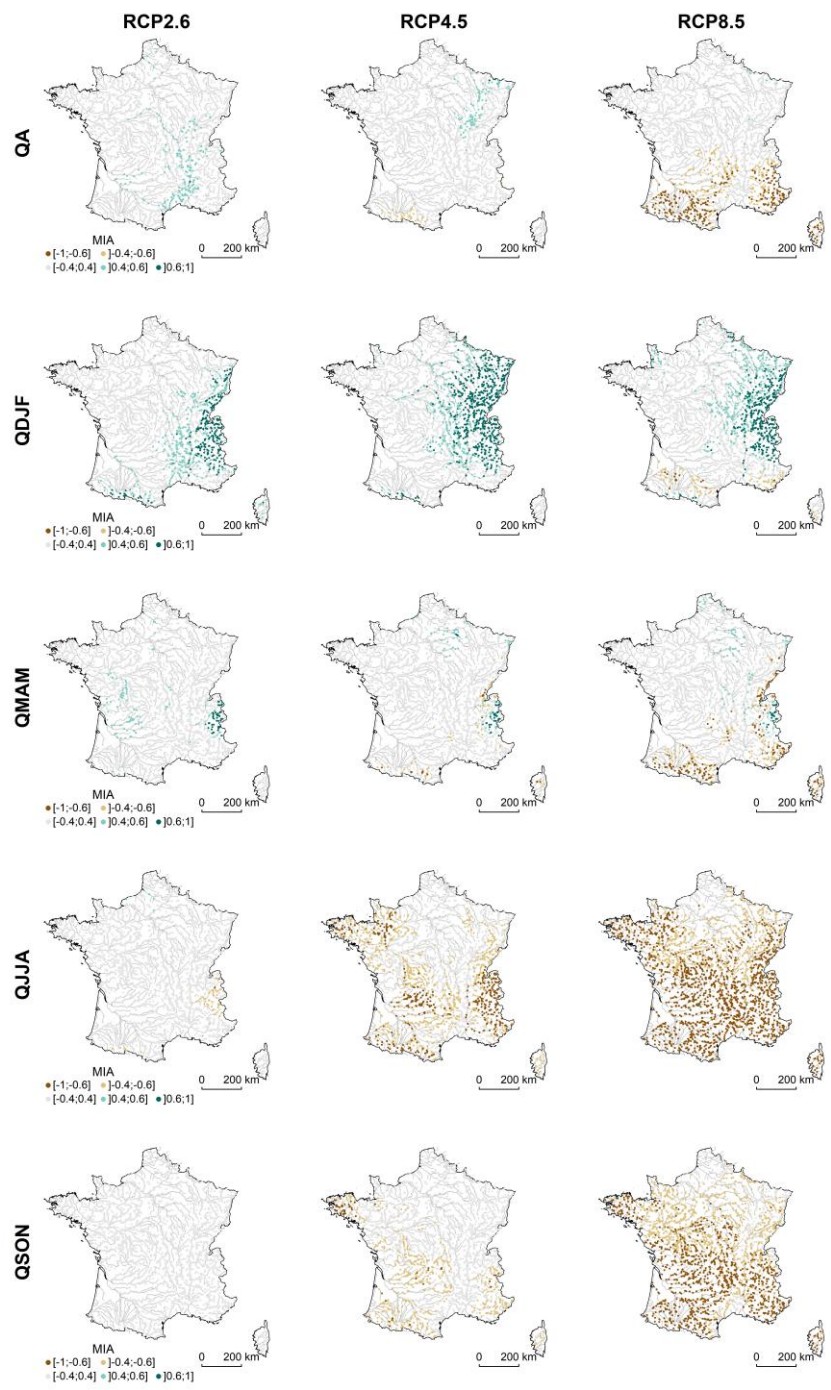

**Figure 8: Agreement in median change in the annual flow QA and the seasonal flows QDJF, QMAM, QJJA, and QSON for the three RCPs.**





The increase in mean winter flows is predominant across France for the three RCPs, except in the southern France where they decrease for RCP8.5. Median changes between the reference period and the two time slices are limited for most of France (Fig. 7). The most significant and positive changes with an agreement between projections can be found for *QDJF* in the Alps and Pyrenees (Fig. 8) for the three RCPs. In these mountainous areas, as a consequence of the increase in air temperature (even moderate for RCP2.6), a change in the phase of precipitations is projected, with snow packs forming later and melting earlier

for the three GHG emission scenarios, and finally low flows in winter will be less severe. On average, catchments with a mean elevation above 2400 m.a.s.l. display median changes between 25 % and 110 % under RCP4.5, and between 50 % and 180 % under RCP8.5, respectively. There are also agreements on changes in *QDJF* in the south of France, but these indicate a decrease. However, the extension is rather limited. For RCP8.5, a strong contrast is observed in the south-west: winter flows are likely to increase in the mountainous headwaters, while hydrological projections do agree on a significant decrease in the

lowlands.

As for mean winter flows, significant increases in spring flows are projected in the Alps and the Jura for the three RCPs, while decrease is projected with moderate to high agreement in southern France for RCP8.5. Spring is certainly the season with the weakest changes ($IQR \subset$ [-15; 15] % whatever the RCP) and with the less agreement on changes. There is less agreement in hydrological projections for mean spring flows than for mean winter flows (e.g. 8 % of the simulation points for *QMAM* against

16 % for *QDJF* under RCP4.5 again). As previously outlined, the increase in mountain flows can be attributed to a change in the dynamics of snowpack melt, driven by a decrease in snowfall and an earlier snowpack melting.

Under RCP4.5, the values of the median changes between the reference period and the end of the century are around +14 % and +7 %, respectively, for mean winter and spring flows. Under RCP8.5, these values are of the order of +11 % and +6 %, respectively, for mean winter and spring flows.

Mean summer and autumn flows are mostly down throughout mainland France for both moderate to high emission scenarios RCP4.5 and RCP8.5. Furthermore, the pattern of median changes for *QJJA* and *QSON* under RCP4.5 and RCP8.5 look similar. Changes in summer and autumn are structured along a north-south axis, with the greatest decreases in the south-west (between -63 % and -32 % in the Garonne basin compared with between -13 % and +2 % for the coastal rivers along the Channel for mean summer flows under RCP8.5. Summer is certainly the season with the most significant changes and with the highest

agreement on these changes. The agreement is present over almost 20 % and 65 % of mainland France in summer under RCP4.5 and RCP8.5, respectively. The northern of France is the only area where hydrological projections do not agree on the sign of change and where the median changes are limited. Similarly to the changes projected in summer, the median changes are almost entirely downward in autumn, reflecting a decrease in water availability in this season and hence a gradual extension of low-flow periods from summer to autumn throughout the 21[st] century. The agreement is projected for near 45 % of the

simulation points and these points are located in southern France under RCP8.5.





| Variable | RCP2.6 | | RCP4.5 | | RCP8.5 | |
|---|---|---|---|---|---|---|
| | 2041-2070 | 2070-2099 | 2041-2070 | 2070-2099 | 2041-2070 | 2070-2099 |
| Annual flow | 1.7; 5.0 | 6.2; 9.8 | -1.7; 4.4 | 0.0; 7.4 | -1.7; 6.3 | -10.1; 4.6 |
| Winter flow | 2.7; 13.1 | 7.6; 13.9 | 5.4; 15.8 | 9.9; 18.9 | 6.6; 18.8 | 2.7; 19.9 |
| Spring flow | 1.3; 5.9 | 4.2; 12.5 | -3.5; 5.8 | -0.6; 12.0 | 0.8; 10.0 | -6.5; 11.9 |
| Summer flow | -0.9; 5.6 | 1.2; 8.5 | -20.7; -12.7 | -17.5; -6.5 | -17.1; -5.6 | -38.1;-21.4 |
| Autumn flow | -8.9; 2.2 | -1.3; 6.4 | -13.1; -2.8 | -18.8; -8.5 | -22.1; -11.0 | -37.4; -20.5 |
| Annual recharge | 5.5; 11.4 | 8.3; 13.5 | 0.5; 7.8 | 4.3; 14.0 | 3.8; 11.8 | -3.0; 12.0 |

**Table 6: Interquartile range of the median changes between the reference period 1976-2005 and the two time slices 2041-2070 and 2070-2099.**

## 7. Discussion

Most of the maps show a north-south gradient. This gradient is already present in the ensemble of climate projections and
particularly pronounced at the end of the century for the high emission scenario. The ensemble of projections suggests:

- An average warming projected between +3 °C and +5.5 °C, with more significant change in mean annual temperature of around +1.5 °C in the south-east of France compared with the north-west,
- No consensus on the sign of the changes in annual precipitation except for the north-east of France (increase), and the south-east and south-west (decrease),
- An increase in winter precipitation between +10 % and +45 %, more pronounced in the north of France and lower, or even uncertain, in the south,
- A decrease in summer precipitation across France, particularly marked in the south-west of France.

As expected, for the winter and spring seasons, the ensemble of projections points towards an increase of water resources in
the north-east and a decrease in the south-east and south-west of France. For the summer and autumn seasons, the majority of
France experiences a decrease in water resources and the north part of France is less affected than the south part. The rivers of
the lowlands in south-western France, as well as the small Mediterranean coastal rivers, appear to be hot spots, with an almost
general decline in all seasonal flows.

Unsurprisingly, global warming, even moderate, impact the dynamic of snow pack in the Alps and the Pyrenees, demonstrating
the high sensitivity of the basins in the mountainous areas. Rivers influenced by snow under current conditions are likely to





experience changes in the magnitude and temporal pattern of low flows with an increase in winter flows and a decrease in summer flows and shifts of the flow peak from spring to winter.

Although France as a whole is expected to dry out in summer, this trend is more uncertain in northern France. In northern France, confidence in the sign of the change in summer is low due to offsetting effects. In this region, river flow regimes are primarily influenced by large aquifers, which regulate flows, with a significant contribution of groundwater to low flows during 535 summer months. By the end of the century, there will be a balance between increased winter recharge, which may contribute to lower flows, and higher evapotranspiration and lower rainfall in summer. In addition, the models are not as effective in representing groundwater-river exchanges (as previously mentioned in Section 5.2.1).

The magnitude and the spread of changes in summer and autumn over France are correlated with the projected level of GHG emission: 80 % of the simulation points show median changes in *QJJA* between -47 % and -15 % (median: -30 %) under 540 RCP8.5 compared with -23 % to -2 % (median: -12 %) under RCP4.5, and 80 % of the simulation points show median changes in mean autumn flows between -44 % and -15 % (median: -28 %) under RCP8.5 compared with -23 % to -4 % (median: - 13 %) under RCP4.5. There is significant overlap between the intervals for the changes in winter and spring.

The ensemble of future streamflow projections produced within the Explore2 project is certainly the richest transient multi-scenario and multi-model ensemble developed in France. This dataset allows for the first time investigating uncertainties in 545 future river flow regimes considering six uncertainty sources: RCPs, GCMs, RCMs, BCs, HMs, and the climate internal variability (Evin *et al.*, in prep). Uncertainties under RCP8.5 high-emission scenario at the end of the century are here illustrated with changes projected by the HMs forced by the four climate story-lines. The results displayed for the four storylines (Fig. 9, Fig. 10) do confirm that these specific forcings lead to contrasting hydrological futures and that they are useful for making stakeholders aware of uncertainties on potential recharge and streamflow. Projected changes in annual potential recharge and 550 winter and summer flows are well correlated with the main characteristics of the climate projections:

- The green storyline is the wettest one: Flows mostly increase in winter (*IQR* = [-3, 32] % across France), except in the most western and southern parts of France, and, although this storyline suggests the highest increase in winter precipitation, only a limited area in northern France may experience less severe low flows (*IQR* = [-29, -1] % across France),

- The yellow storyline is the less extreme one: The spatial patterns are similar to the ones obtained with the green storyline, but with less contrast in winter (*IQR* = [7, 26] % across France in winter),

- The purple storyline is the most contrasting one: increases in potential recharge and winter flows are locally projected in the north-eastern part of France in response to more abundant winter precipitation, while a significant decrease in these two metrics is predicted in the south-western part of France (*IQR* = [-21, 15] % across France), basins in south-560 west France are particularly affected in summer by the reduction in recharge (*IQR* = [-56, -36] %),



- The orange storyline is the driest one, leading to a generalised decrease in flows whatever the season (*IQR* = [-25, -3] % across France in winter, *IQR* = [-66, -53] % across France in summer).

Not surprisingly, the summer flows and, to a lesser extent, the autumn flows (not shown), are projected to decrease, and the winter flows are expected to increase in mountainous areas, regardless of the storyline.




**Figure 9: Changes in the annual recharge and the winter and summer flows between the benchmark period and the end of the century for the four storylines. The areas coloured black on the maps are those where there is no aquifer and where the notion of recharge is not relevant. Median changes are shown for seasonal flows based on available HMs.**





**570**

**Figure 10: Changes in annual temperature, precipitation and streamflows for three large catchments in France and for RCP8.5 (solid thick lines are moving average over 20 years, individual hydroclimate projections are displayed in grey).**

## 8. Conclusions

A nested multi-scenario, multi-model approach to understand future certainty and uncertainty, and to access local climate and
**575** catchment scales, was used to produce an updated nationally consistent multi-model ensemble of transient hydrological projections at the daily time step. This ensemble results from the combination of three GHG emission scenarios, a set of 17 climate modelling chains GCM-RCM and two bias correction methods, and 9 surface hydrology models and 4 groundwater hydrology models (one to simulate groundwater recharge and three to simulate groundwater level).



A comprehensive evaluation analysis was performed here using a large set of time series of daily streamflow and groundwater levels. The diagnostic provides an overview of the quality/performance of surface hydrology and hydrogeology models, using the SAFRAN reanalysis as input data. Modest HMs performance is more often found in the north, where aquifers regulate the dynamics of river flows and in the mountainous areas where surface HMs have to account for snowpack. The ability of the entire modelling chain to represent observed river flows when HMs are fed by historical runs was assessed by comparing two flow quantiles and the mean annual streamflow extracted from the observations and the simulations over the baseline period
1976-2005 and was deemed adequate. The magnitude of changes in recharge and streamflow is not uniform but organized along a north-south axis, and depends on the changes in climate and on the geological characteristics of the areas. Unsurprisingly, climate change will increase the severity of low flows in summer and decrease the severity of low flows in winter. Most of the rivers in south-western France and along the Mediterranean coastal will be hot spots, with an almost general decline in all seasonal flows.

To our knowledge, the Explore2 ensemble has used an unprecedented diversity of climate models (compared to the UK (eFLaG) dataset (Hannaford *et al*., 2023), which is based on a single GCM and a single RCM) and hydrological models (compared to the Australian ensemble produced by Zheng *et al.* (2024)). In addition to the technical challenge of producing such an ensemble, the richness of the ensemble lies in its use for operational and research purposes, for example: the ability to characterise the uncertainties as described, but also the ability to produce consolidated results in the case of strong agreement
between the projections, and to step back from the results in the case of disagreement, thus promoting understanding of the divergences between hydrological projections.

An immediate follow-up will be to analyse results of the Explore2 dataset on the basis of global warming of 4 °C above pre-industrial levels. The scenario of +4 °C by the end of the century, included in the TRACC (Trajectoire de Réchauffement de référence pour l'Adaptation au Changement Climatique, reference trajectory for adapting to climate change), is the reference
warming scenario chosen by the French government for the French National Adaptation Plan for Climate Change (PNACC-3) launched in 2024. All planning documents and adaptation strategies in continental France will have to be based on this trajectory.

**Author contributions**

ES and GE led the study. The selection of reference gauging stations for the evaluation of surface HMs was carried out by LS,
GT and ES. The selection of reference piezometers for the evaluation of groundwater HMs was carried out by JPVe. Hydrological simulations were run by JPVe (AquiFR, EROS), SM (CTRIP), LS (GRSD), JB (J2000), RA (MONA), MLL and JG (MORDOR-SD), CM (MORDOR-TS), PH (ORCHIDEE), OR (RECHARGE), FR (SIM2), FC (SMASH). LH, JPVe, JPVi, AJ and ES led the evaluation. All the authors contributed to the delivery of the dataset, read and approved the final paper.



**Data availability**

The Explore2 dataset is associated with the following digital object identifier https://doi.org/10.57745/JJWOYS. The hydrological data can be downloaded in netCDF file format through the open platform for French public data dedicated to the Explore2 project (https://entrepot.recherche.data.gouv.fr/dataverse/explore2) and the DRIAS-Eau website (https://www.drias-eau.fr/). The Explore2 dataverse is the storage location for the technical documentation of the Explore2 project (written in French). The MEANDRE portal (https://meandre.explore2.inrae.fr), developed within the LIFE Eau&Climat project, is an

interactive visualisation platform for Explore2 main results (including Explore2 key messages on these three aspects of changes in future average flows, low flows and high flows, maps and time series of any individual hydroclimate projection or any combination of these).

**Acknowledgment**

Explore2 was funded by the French Ministry of the Environment and the French Biodiversity Agency (OFB). The total grant

was 979 k€. The hydrological projections with the ORCHIDEE and RECHARGE models were produced using HPC resources from respectively GENCI-IDRIS and from the cluster at the Centre de Calcul Scientifique en région Centre-Val de Loire, and PH was supported by Explore2 and the Institut Pierre Simon Laplace (IPSL).

The MEANDRE and DRIAS-Eau portals were developed within the LIFE Eau&Climat project. The LIFE Eau&Climat project (LIFE19 GIC/FR/001259) received funding from the European Union's LIFE programme. It was co-funded by the French

Water Agencies, ADEME, and the Rhône-Alpes Region.

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
