# Peer review of "A large transient multi-scenario multi-model ensemble of future streamflow and groundwater projections in France"

_EGUsphere, 2025_

## Author Comment (AC1)

**We would like to thank the two reviewers who kindly provided us with their comments.**

**The initial objectives described in the introduction were to present "the methodology underlying the dataset, the evaluation of the hydrological models against daily streamflow and groundwater level observations, the assessment of the future streamflow and recharge projections, the data availability and the ways of accessing the data and understanding the results (mainly through visualisation tools)". These objectives were too ambitious for one paper. It is clearly unsatisfactory and the content of the different sections is finally unbalanced. The article is already too long. Based on the recommendations of the two reviewers, we have chosen, in the next version, to reorganize the paper, to complete the sections on the model evaluation procedure and the description of the results, to add a literature review and to reduce the section devoted to the transfer of data and project results to stakeholders. The transfer to users will be the subject of a future, much more comprehensive article (including visual tools, the technical documentation and the interpretation of summary sheets).**

**We will pay more attention to the numbering of tables, figures and sections in the next version. This poor numbering made the manuscript difficult for reviewers to read. We apologise for any inconvenience caused and undertake to rewrite the sentences (deemed too long) and avoid redundancies.**

**RC1**: 'Comment on egusphere-2025-1788', Anonymous Referee #1, 17 Jun 2025

General comments

Sauquet et al. presented in their manuscript the results of French national project called Explore2 to provide hydrological climate projections in France for streamflow and groundwater changes until the end of this century. It is a successor project of Explore2070 which took place in in the early 2010ies. The motivation to update projections were to extend projections to the end of the century with newer climate model data and to increase the locations for projections. Many institutions were involved over several years from science, administration and stakeholders in water management. The objective of the project also to provide a better accessibility and understanding of the data for stakeholders compared the predecessor project. The main objectives of the manuscript were to describe the ensemble of the simulations, to evaluate the ensemble in past climate and present main results of the climate projections.

They used a subset from the EURO-CORDEX CMIP5 climate chain ensemble, two bias correcting methods, nine surface hydrology models and four groundwater models. Models were evaluated in the past both using observed forcing and climate model data at large number of observation location, i.e. > 600 gauging stations for surface hydrology models > 200 piezometers for groundwater models over many years. Some of the models were calibrated using similar data as for evaluation. Not all models cover whole France. The authors present evaluation under multiple aspects and mention thresholds for acceptable performance.

The projections cover up to approx. 4000 simulation points for streamflow projections covering whole France and parts of France for groundwater recharge projections. Results for evaluation and projections focused on the surface hydrology models as results of the groundwater models are (or will be) presented elsewhere. However, there is a sufficient overview of the groundwater part presented in this manuscript.

The substantial content of the manuscript is, however, not matched with the quality presenting the data. Most importantly I miss a literature review on the current knowledge of hydroclimatic projections

in France (and Europe). The question must be answered, to my opinion, how the presented results embed in and expand existing knowledge.

The second largest issue is the readability of the manuscript, which suffers under many aspects. Most problematic are the large number of the results, the imprecise language which requires the reader to interpret what could be meant and the manuscript structure. To my opinion the manuscript will largely improve with using a commonly used structure to separate data/methods and results/discussion. I will detail the arguments below. With this impression I recommend to major revisions for this manuscript.

Detailed comments

Literature review

As mentioned above the manuscript is not largely referring to previous work. This should also include regional studies. This aim was mentioned in the overview of the manuscript to be part of section 7, but I could not find a relevant discussion with previous literature. This literature review should also include model decisions (e.g. the impact of bias correcting methods …). Section 7 rather adds new own results not presented before.

➔ *A review of the literature on impact studies concerning France has been published in French (https://doi.org/10.57745/XKMLMG). This is a part of the technical documentation of the Explore2 project. We will summarise the content of this report in the introduction and compare the results obtained here with the conclusions of the identified previous studies.*

Readability

First is the large number of results. The objective was to show only main results, and I think this can be more condensed. The large number of results in the figures and tables is supplemented with additional numbers in the text, which makes it difficult to read (e.g. on page 28). Generally, I advise that these numbers in the text could be presented in a table, however, given the need to condense results for this manuscript, I would vote for a more qualitative text description to put the chosen results in tables and figures into value.

➔ *We will reduce the number of Tables (e.g., Table 1 is redundant with Fig. 1). We will follow the reviewer's advice by removing the numbers from the text to make the article easier to read.*

Second is the imprecise language. As it is a complex data set covering many dimensions it is essential to be precise. The reader may be able to follow what is meant but it requires a very thorough reading. A table and a figure, however, should be self-explanatory to a certain extent. This is often not given. As an example for many other parts is the presentation of Table 6. The unit is not provided, so one cannot say if the numbers are percentage changes or absolute changes. It can only be assessed from the context that the "interquartile range of the median change …" in the caption description refers to an interquartile range between simulation points (space) and the median change to a set of simulations varying in climate models, bias corrections and hydrological models.

➔ *We have sought to clarify the nature of the statistics (overall, spatial, etc.). We need to improve the text in this regard.*

Additionally, I could often not follow what the authors interpret from figures, either by wrong references to figures or that they were imprecise with mentioning patterns or regions. For example, in lines 482-483 "The most significant and positive changes with an agreement between projections can be found for QDJF in the Alps and Pyrenees (Fig. 8) for the three RCPs." I could follow this statement

with looking at Figure 7. Similarly, I could not follow the impressions of the elasticity values (lines 338-342) shown in Fig. 3: Are the values for temperature good, or for precipitation in the summer?

→ *Please accept our apologies for any errors in the references to figures, sections and tables.*

Manuscript structure

The reading flow is often interrupted by switching between methods and results as the manuscript is more structured by content (evaluation, future projections …) than separating data/methods and results/discussion. For example, the results on projected changes start with a description of a multi-model index of agreement (Eq. 2). Ideally the manuscript would keep the content structure but replicate this structure for the data/methods section and the results/discussion section.

→ *We have reorganised the manuscript in accordance with these recommendations. The data used for the hydrological modelling will be presented in the first two sections (first: observations and meteorological reanalysis, and then: climate projections). The section on the co-construction process with stakeholders will be shortened and incorporated into the conclusion. Methods and results will be presented in different sections.*

Focus of the manuscript

In the abstract the authors mention prominently that they want to present the data availability and understanding the results through visualisation tools. This part is in the manuscript rather short. If the authors want to present their data availability, the understanding thereof and their visualisation tools as promised in the abstract I think this needs to be more elaborated in the manuscript.

For example, the fact sheet in the supplemented is quite complex and is not put in value in the manuscript text. I would have liked to see more details on the description of these sheets and how these sheets help to make informed choices pre-select hydrological models (mentioned in section 5).

The data availability is only presented with a link (section Data availability). There is no guidance in the manuscript or on the webpage how I can get netCDFs of a certain figure for example. The webpage is available in an English version but 99% of the text is still in French. Besides the language barrier (which can be solved with translation tools nowadays) the data is hidden in multiple links.

There is a link for each model for each RCP for each subregion, which leads to a new webpage with new links with filenames, which relate to different climate chains and bias correction methods. Similarly, the visualisation tool is presented just with a link but not put in value in the manuscript In a similar note: In the conclusion in line 595 the authors advertise their data set with the ability to step back from the results in case of disagreement and promoting an understanding of the differences between hydrological models. This is indeed possible with such a data set, but it is not presented in the manuscript. The authors show mainly median results between hydrological models (except of a minor subtopic with Figure 5). In Table 3 there are thresholds presented for model performance, but nothing mentioned about the consequences. It would be interesting if the authors would include a detailed description of their fact sheets and how one could potentially use them.

→ *Based on the two reviewers' comments, we understand that we were overly ambitious in attempting to present both the Explore2 dataset and the main findings, as well as how stakeholders can use the information provided by the project (including model selection). The transfer of knowledge from the project will be the subject of a separate article. Thus, the formerly section entitled "Co-construction between scientists and stakeholders" will be shortened and moved to the conclusion section.*

Minor points

Reply
- In the abstract there are no results summarized for the groundwater models
  ⇨ *The following finding will be added: "In addition to northern France, annual groundwater recharge is projected to increase slightly in the north-east and remain unchanged elsewhere by the end of the century, according to the RCP8.5 scenario."*

- Section overview is inconsistent with section headlines and content (lines 91-95). Similarly, reference is probably wrong in line 446 (Table 7 is not existent), and section references (line 426).
  ⇨ *Sorry again for the wrong numbering.*

- Line 103: What are the criteria selecting the subset of climate models? The cited study, which could answer the question is only available in French.
  ⇨ *The criteria used for selecting GCM/RCM are related to data availability, model diversity and simulation quality:*
  *● Availability and consistency of RCPs: each pair must be available for at least two RCP scenarios,*
  *● Quality of GCMs: GCMs must be considered realistic for Europe (McSweeney et al., 2015),*
  *● Consistent distribution: the spread of the climate change signal from the EURO-CORDEX ensemble must be preserved,*
  *● Consistency with CMIP6 ensemble: changes under RCP8.5 were compared to those projected by the CMIP6 ensemble under the SSP5-8.5 scenario for the end of the century and for the two contrasting seasons DJF and JJA. The objective of the comparison was to exclude projections out of the range of the CMIP6 ensemble for both temperature and precipitation changes in winter and summer,*
  *● Evolving aerosol forcing: include the latest EURO-CORDEX simulations allowing evolving aerosols,*
  *● Redundancy : enable the use of the QUALYPSO method for uncertainty decomposition (Evin et al. 2021), which requires each GCM and RCM to be present more than once.*

- Line 111: Why is the reference period (sometimes called with different names, try to use consistent naming for better readability) chosen to be 1976 to 2005 opposite to the latest AR5 and AR6 reports? This could be better motivated.
  ⇨ *The term "reference period" has been replaced by "reference period" throughout the text to ensure consistency in terminology. 2005 is the last year in the historical runs for the EURO-CORDEX CMIP5 ensemble, which is why we have taken the period 1976-2005 as a reference.*

- Line 111: The meteorological forcing is on a 8 km grid, but small catchments up to 64 km2 (line 206) are included. Given that the effective resolution of the climate model is > 12 km and that of SAFRAN also > 8 km, which is more reflected by the underlying station distribution, I have the impression that too small catchments are included in the study. Can you comment on this?
- Table 2: Can you include the model resolution in Table 2? Right now, I need to look in the supplement. How did you bridge the gap from 8 km resolution after bias correction to the much finer hydrological model resolution?
  ⇨ *Information regarding the size of the catchment is provided in line 207, "Given the 8-km spatial resolution of the gridded climate projections that were used as forcings for hydrological modelling, points with a drainage area of less than 64 km² have been excluded". We will add the following sentence in the section entitled "Surface hydrographic*

*network and actual simulated points": "The minimum size of simulated catchments is 64 km², in accordance with the spatial resolution of climate projections". We will add the distribution of the drainage area of the simulated points for each surface HM.*

- Fig 5: how do the results differ spatially?
  ⇨ *Additional calculations will be presented and commented.*

- 1 shows differences in bias correcting methods. This should be mentioned. As they are quite substantial, I would also like to see (despite condensing results) a graph showing not only a median change over all simulations (Fig 6), but also split into bias correction methods. Figure 5 would also be interesting to see and discussed for the bias corrections separately.
  ⇨ *We will produce the same graph with the results of the CDF-t bias correction (BC) method as in Figure 5. Maps can be derived from the hydrological projections obtained with CDF-t. It should be noted that not all HMs used the climate projections obtained with CDF-t as inputs. The comparison will therefore not be complete (it will only be based on the GRSD, SMASH and MORDOR-SD models available at national level, see Table 2). It should be also noted that the uncertainty due to BCs in the Explore2 climate projections was found to be low, as the BCs applied are two similar approaches based on quantile mapping.*

- Table 3 (I refer to the second Table 3): Can you include formulas for the metrics (maybe in the assets or provide code)?
  ⇨ *The formulas will be detailed in the supplementary materials.*

- Figure 3: some metrics are not mentioned in the text (aCDC, alphaQA). Consider to put them in value or delete the figure.
  ⇨ *These maps will be deleted.*

- Line 259 & 374: What do the authors mean with "historical runs", I assume hydrological runs forced with climate model data from the period xxx to xxx?
  ⇨ *To clarify the text, we will add the following sentence "Hydrological runs are simulated data obtained by HM forced with bias-corrected projections for the reference period 1976-2005".*

- Line 389: term "expected" is misleading as this metric can take on any value.
  ⇨ *We suggest this new sentence: The $R_{RH}$ criterion should take a value between -1 and +1 if no significant additional bias appears with the use of bias-corrected climate data.*

- Line 392: Which are the 17 runs? The owns with RCP8.5 only?
  ⇨ *A set 17 pairs GCM/RCM were used to build the climate projections. Historical runs are common to the projections based on the same pair GCM/RCM. This will be added in the section on climate projections.*

- Line 438: title of section is misleading as also results mid of century are presented
  ⇨ *The title is now: "Summary of the projected changes on mean flows and mean groundwater recharge"*

- Lines 439-440: Starting a results paragraph with saying that parts of the results are presented elsewhere reads strange.
  ⇨ *This sentence will be moved to the next paragraph.*

- Line 481: What do the authors mean with changes are limited when referring to Fig. 7? I can see substantial changes from ranging from minus to plus 50% covering most and parts of France, respectively?
  ⇨ *The changes in winter are limited compared to those projected for summer (see Table 6). Using the term "limited" is not relevant. We will remove this sentence, as it does not provide any additional information (details are provided later in the paragraph).*

- In whole section 6: I would recommend not to use the wording "significant change" as no statistical test was applied
  ⇨ *The reviewer is absolutely right. We will use "major changes" instead.*

- Line 504: The authors say that the most significant changes are in the summer. Looking a Fig. 7 RCP8.5 I do not see a large difference to fall.
  ⇨ *The decreases in summer are slightly greater than those projected in autumn (a difference of a few %). This small difference suggests that the changes observed in summer and autumn should be described in similar terms.*

- Line 514: North-south gradient. Please be a bit more specific to which season and which variable as I do not agree on the sentence "most maps show a north-south gradient".
- Line 514: "This gradient is already present in the ensemble of climate projections". Please add that this was not shown in the manuscript or put a figure in the Supplement.
  ⇨ *We chose to use the same colour palette for all maps, firstly adapted to display the changes predicted for the end of the century in the RCP8.5 scenario. This highlights this north-south gradient at the end of the century, but not necessarily for other scenarios and time slices. Spatial contrasts will be detailed using results obtained for two basins (one in the north and one in the south). Furthermore, in order to better understand the reasons for spatial contrasts, maps illustrating changes in precipitation will be added to the supplementary materials.*

- Line 546: Please mention that with the storylines only climate model uncertainty is presented (and maybe only a part), while the other sources are presented elsewhere.
  ⇨ *We will add this comment when presenting the storylines.*

- 9: Do you want to discuss why the yellow story line (which should be a moderate one) is more extreme than the median changes presented in Figure 7? This is pretty obvious by design, but I recommend the authors to do not leave these kind of thoughts all readers.
  ⇨ *The comparison is not straightforward between Figure 7 and Figure 9, as the colour palette is not the same. We will harmonise the palettes and compare the median change maps for the RCP8.5 scenario with those obtained for the four storylines.*

- Fig. 10 is not mentioned in the text.
- Fig. 10: I would suggest plotting also all individual projection.s with moving averages. The annual variability is not the point here to my opinion.
  ⇨ *Fig. 10 will be deleted in the next version. The results shown in Fig. 10 will be used to support, in the text, the pattern of changes (north-south gradient).*

- Line 610 points to an outdated data set, there is a new doi provided
  ⇨ *The sentence will be modified: "The Explore2 dataset is associated with the following digital object identifier https://doi.org/10.57745/YHMBHC"*